# A Single Nucleotide Polymorphism Translates into a Radical Amino Acid Substitution at the Ligand-Binding Site in *Fasciola hepatica* Carboxylesterase B

**DOI:** 10.3390/genes13101899

**Published:** 2022-10-19

**Authors:** Estefan Miranda-Miranda, Silvana Scarcella, Enrique Reynaud, Verónica Narváez-Padilla, Gisela Neira, Roberto Mera-y-Sierra, Hugo Aguilar-Díaz, Raquel Cossio-Bayugar

**Affiliations:** 1Departamento de Artropodología, Centro Nacional de Investigaciones Disciplinarias en Salud Animal e Inocuidad, Instituto Nacional de Investigaciones Forestales Agrícolas y Pecuarias (INIFAP), Boulevard Cuauhnahuac No. 8534, Jiutepec 62574, Morelos, Mexico; 2Laboratorio de Biología Celular y Molecular, Centro de Investigación Veterinaria de Tandil (CIVETAN), CONICET, UNCPBA, Tandil 7000, Argentina; 3Departamento de Genética del Desarrollo y Fisiología Molecular, Instituto de Biotecnología, Campus Cuernavaca, Universidad Nacional Autónoma de México, Avenida Universidad, 2001, Apartado Postal 510–3, Cuernavaca 62210, Morelos, Mexico; 4Centro de Investigación en Dinámica Celular de la Universidad Autónoma del Estado de Morelos, Campus Cuernavaca Morelos México, Av. Universidad 1001, Chamilpa, Cuernavaca 62210, Morelos, Mexico; 5Centro de Investigación en Parasitología Regional (CIPAR), Universidad J.A. Maza, Guaymallen 5519, Argentina

**Keywords:** SNTP, carboxylesterase, amino acid substitution, *Fasciola hepatica*, anthelmintic resistance, bioinformatics

## Abstract

*Fasciola hepatica* anthelmintic resistance may be associated with the catalytic activity of xenobiotic metabolizing enzymes. The gene expression of one of these enzymes, identified as carboxylesterase B (CestB), was previously described as inducible in adult parasites under anthelmintic treatment and exhibited a single nucleotide polymorphism at position 643 that translates into a radical amino acid substitution at position 215 from Glutamic acid to Lysine. Alphafold 3D models of both allelic sequences exhibited a significant affinity pocket rearrangement and different ligand-docking modeling results. Further bioinformatics analysis confirmed that the radical amino acid substitution is located at the ligand affinity site of the enzyme, affecting its affinity to serine hydrolase inhibitors and preferences for ester ligands. A field genotyping survey from parasite samples obtained from two developmental stages isolated from different host species from Argentina and Mexico exhibited a 37% allele distribution for 215E and a 29% allele distribution for 215K as well as a 34% E/K heterozygous distribution. No linkage to host species or geographic origin was found in any of the allele variants.

## 1. Introduction

*Fasciola hepatica* is the causative agent of a global zoonotic disease known as fasciolosis [1], a neglected helminthiasis affecting the liver of several livestock species as well as humans [2]. This liver parasite causes losses of billions of dollars in the meat, wool, and milk industry and an even a larger amount invested in public health expenses destined to treat millions of humans that suffer fasciolosis around the world [3]. Antiparasitic compounds against *F. hepatica* include benzimidazole derivatives such as triclabendazole (TCBZ), a livestock anthelmintic that has been used for several decades as a preventive drug for the control of parasitic flat worms including *F. hepatica* [4]. Inevitably, the indiscriminate use of TCBZ in livestock led to liver fluke resistance and forced livestock producers to use more expensive treatments, opening the door to multiple anthelmintic resistance [5].

The liver fluke is only one example of a general tendency of parasitic helminths to evolve anthelmintic resistance by neutralization of anthelmintic compounds toxicity on the parasites [5,6]. This biochemical scenario may involve the active role of xenobiotic metabolizing enzymes (XMEs) which are known to have an active role in the drug resistance against the antiparasitic effects of man-fabricated chemicals reported in many parasite species [6].

Carboxylesterases are XMEs commonly found in most eucaryotic species [7]; these enzymes catalyze the break-down of small molecules that have carboxylic acid ester, amide, and thioester functional chemical groups into reaction products containing alcohols and organic acids [8]. The expression and specific enzymatic activity of carboxylesterases play a significant role in the resistance mechanisms shown in a variety of parasites against antiparasitic chemicals, such as the paradigmatic resistance of *Plasmodium falciparum* to antimalarial compounds [9,10]. Carboxylesterases are also proven to be responsible for insecticide resistance in blood-sucking insects such as the mosquitoes *Anopheles funestus* [10], *Culex nigripalpus* [11], and the kissing bug *Rhodnius prolixus* [12].

Adult *F. hepatica* carboxylesterase type B (CestB) expression was found to be induced under TCBZ treatment of experimentally parasitized sheep [2]; this serine hydrolase exhibited catalytic enzymatic activity on the synthetic chromogenic substrate α-Naphthyl acetate (ANA) and monomeric and homodimeric isoforms of CestB were found in crude extracts and cytosol protein fractions using SDS-PAGE zymograms [3]; ANA esterase chromogenic substrate may be also useful for ligand-docking 3D modeling by dedicated algorithms [13,14]. The CestB gene transcript nucleotide sequence codes a 2205 bp open reading frame that translates into a 735 amino acids protein found to be an integral component of the cellular membrane by gene ontology enrichment analysis (GOEA) [1]. Further bioinformatics analysis indicated that orthologs of this enzyme in other species of helminths are implicated in drug metabolism and interact with other enzymes that are known to metabolically modify drugs [3]. The goal of this study is to identify single nucleotide polymorphisms (SNPs) on the CestB gene obtained from the gene and protein databanks as well as from DNA obtained from parasite isolates and to make a bioinformatic assessment of the probable impact of the identified alleles on the structure and enzyme functionality over anthelmintic metabolites.

## 2. Materials and Methods

### 2.1. Protein Sequences

UniProt amino acid sequences A0A8A1L7B4 and A0A4E0S0J7 and their respective Alphafold 3D model in PDB files were downloaded from www.uniprot.org (accessed on 10 September 2022) [14] and https://alphafold.com (accessed on 10 September 2022) [15]. Additionally, complementary GenBank amino acid sequences MT843326, MW655750, D915_000180, and THD28967.1 were obtained from www.ncbi.nlm.nih.gov (accessed on 10 September 2022).

### 2.2. Protein 3D Modeling

Protein 3D modeling was performed by using the Mol* online algorithm [16] (https://molstar.org) (accessed on 10 September 2022). Ligand docking and ligand binding site identification and 3D modeling of the ligand-binding site were performed by the Coach-D online algorithm [17,18] https://yanglab.nankai.edu.cn/COACH-D/) (accessed on 10 September 2022).

### 2.3. Parasite Material

Adult *F. hepatica* were recovered from the bile ducts of bovine, ovine, and equine hosts raised in Argentina and Mexico. The parasites were rinsed extensively with sterile phosphate saline solution (PBS; NaCl 137 mM, KCl 2.7 mM, Na_2_HPO_4_ 10 mM, KH_2_PO_4_ 1.8 mM) pH 7.2 at 37 °C to remove bile and/or adhering materials. An egg-free section from the parasites was dissected and preserved in RNAlater^®^ (ThermoFisher, Waltham, MA, USA) until nucleic acid extraction according to a previous report [2]. Fecal samples were collected from cattle-grazing areas and processed for *F. hepatica* egg identification and DNA isolation from 2000 egg aliquots according to a previous report [19].

### 2.4. DNA Extraction

The genomic DNA of each fragment of a single adult parasite sample was isolated following the standard phenol–chloroform procedure [20,21]. Fecal egg DNA isolation was performed on 2000 egg aliquots according to a previous report [19].

### 2.5. PCR Conditions

A set of primers (forward: FExon1CestB 5′-CGGGTCCAAGCAAGGATGAG-3′; reverse: RExon1CestB 5′-CTCTCCTCCGACCATCAAATTC-3′) was designed using the GenBank nucleotide sequence of CestB entries MT843326 and MW655750 and assessed for the amplification of exon one spanning from nucleotide 99 to 1042 in the CestB gene [3] using the Priming design tool algorithm at NCBI (https://www.ncbi.nlm.nih.gov/tools/primer-blast/) (accessed on 10 September 2022). A polymerase chain reaction (PCR) was carried out in 20 μL of total volume containing 4 µL, Promega PCR master Mix 5×, 0.5 µL of each primer at 1 µM, 1 µL DNA 20 ng/µL, and 14 µL H_2_O. The PCR amplifications were carried out as follows: 3 min of denaturation at 95 °C followed by 10 cycles at 94 °C for 15 s, 65 °C for 30 s, and 72 °C for 30 s and programing the thermocycler to subtract 1 °C from each cycle to the annealing step. This was followed by 15 cycles of 93 °C denaturation for 30 s, 60 °C annealing for 30 s and 72 °C extension for 40 s and a final extension step of 72 °C for 5 min. Duplicate PCRs on each individual template DNA were performed to test the reproducibility of the individual DNA bands. In all PCRs, a negative control was included containing the reaction components except for the DNA template.

### 2.6. Sanger Amplicon Sequencing

Amplicons were cleaned by the Wizard Gel and PCR Clean-Up system Promega^®^ and submitted for Sanger sequencing at the IBT-UNAM. The obtained DNA sequences were translated to the respective amino acid sequences by ORFinder at NCBI (https://www.ncbi.nlm.nih.gov/orffinder/) (accessed on 10 September 2022). The Clustal Omega online algorithm was used for a multiple sequence alignment of both DNA and amino acid sequences obtained, and localization of SNP 643 and amino acid substitution at position 215 (https://www.ebi.ac.uk/Tools/msa/clustalo/) (accessed on 10 September 2022).

## 3. Results

### 3.1. Protein Sequence Properties

*F. hepatica* carboxylesterase B nucleotide and amino acid sequences were downloaded from the GenBank and UniProt databases as Fasta files, and links to the respective Alphafold 3D model were used to download their respective 3D model in PDB format. All basic protein properties, such as GOEA and Kegg orthologs, demonstrated that CestB is a serine hydrolase type B, closely related to acetylcholinesterase, neuroligin, and neurexin; it contains a catalytic serine at the active site and is capable of hydrolyzing small synthetic substrates. The enzyme properties, protein domains, and cell location predictions are summarized in Table 1.

### 3.2. Protein 3D Modeling

Alphafold 3D models showed links to identical proteins that were downloaded as PDB files and used for subsequent multiple 3D alignment comparison using the Mol* algorithm, and the results showed a significant structural rearrangement in the overall protein structure (Figure 1).

### 3.3. Ligand-Binding Modeling

The Coach-D algorithm identified the amino acids Lysine 210, Asparagine 211, Glycine 214, Glutamic acid 215, Leucine 216, Valine 217, Glycine 256, Leucine 259, Tyrosine 284, Serine 336, and Threonine 339 at the E215 allele ligand-binding site, whereas the K215 allele identified a ligand-binding site Lysine 215, Tyrosine 258, Isoleucine 340, and Asparagine 735 that were not present in E215 allele. The results are summarized in Table 2.

### 3.4. Ligand Docking Visualization

The ligand-docking 3D models were downloaded as PDB files and visualized in Mol* to identify the amino acids at the catalytic site using the ANA ligand as an indicator of the relative position of the catalytic serine (Figure 2 and Figure 3).

### 3.5. PCR Amplicon Sequencing

The Sanger-sequenced amplicons were analyzed for the SNPs at position 643 and translated to their respective amino acid sequences, and the amino acid at position 215 was identified by Clustal Omega multiple sequence alignment. The results are summarized according to species and geographical origin in Table 3, Figure 4.

### 3.6. Allele Distribution by Species

The PCR-sequenced amplicons obtained from parasitic samples and translation to amino acid sequence at position 215 are graphically represented according to species distribution in Figure 4

## 4. Discussion

Although amino acid 215 is relatively far from the catalytic serine at the core of the enzyme active site, it was identified by the Coach-D algorithm as part of the ligand affinity site (Table 1, Figure 2). The 3D model of both allelic sequences allowed us to identify several structural configurations that affect both the affinity site and overall protein structure, as well as the ligand affinity preference of the two different alleles. The effect of the amino acid substitutions extends to catalytic S336 at the core of the catalytic site, affecting the hydrogen bonds and the hydrophobic interactions between the amino acids at the binding site (Figure 2 and Figure 3).

S336 was identified as the catalytic amino acid of the active site, and 3D modeling indicated that both alleles had a different interaction of the amino acids of the active site with the catalytic serine at position 336. The allele K215 exhibited hydrogen bonds with Glycine 256, Tyrosine 258, Leucine 259, Threonine 339, Isoleucine 340, and notably Asparagine 735, the most distal amino acid in the sequence (Table 2, Figure 4), whereas the catalytic serine in allele E215 exhibited hydrogen bonds with Glycine 256, Tyrosine 258, Leucine 259, Threonine 339, and Isoleucine 340. Remarkably, the interaction of Asparagine 735 at the active site and catalytic Serine 336 disappeared, explaining the affinity shift for small ligand preferences in the catalytic pocket, as summarized in Table 3.

The ligand ANA was always positioned by the modeling algorithm near Serine 336, suggesting that this is the location of the catalytic serine that every serine hydrolase contains within its catalytic site (Figure 4) [22]. This was corroborated by the significant number of hydrogen bonds from Serine 336 to its neighboring AAs (Table 1, Figure 3). A notable difference was the hydrogen bonds of Leucine 259 and Tyrosine 284 that were present only in the E215 allele and the hydrogen bonds of Tyrosine 258 and Asparagine 735 that were present only in K215. These differences may also have affected the observed differences in ligand preferences predicted by the Coach-D algorithm, which showed that among the 21 small ligands tested only two of them shared affinity among both alleles (Table 2). The ligands predicted by the algorithm are well-known irreversible serine hydrolase inhibitors, such as organophosphorus compounds that are widely used in enzyme studies for the identification of different types of serine hydrolases [8]. These families of inhibitors bind covalently to serine at the active site of serine hydrolases, blocking their enzymatic function effectively and irreversibly [23]. Notably, this group of ligands identified by Coach-D includes Sarin, the chemical proposed as a weapon-of-mass destruction during the WWII conflict [24].

A second group of ligands includes choline derivatives analogous to acetyl choline, the most abundant excitatory neurotransmitter in the arthropod central nervous system and the one that induces muscular contractions in mammals [23]. These ligand families and affinities corroborate the results found using gene ontology enrichment analysis, which show that CestB is closely related to acetylcholinesterase, another carboxylesterase B that hydrolyzes the neurotransmitter acetylcholine in the neuromuscular junction, thus permitting muscular relaxation after neurotransmitter release [8]. It is well-known that carboxylesterases, such as CestB, and cholinesterase share affinity to common synthetic enzymatic substrates [25].

Other ligands include small ester synthetic ligands, such as naphthyl acetate (ANA), which are used as esterase chromogenic substrates in histochemical analysis in the laboratory [26]. ANA ligand is a widely used small synthetic esterase substrate in analytical enzymatic kinetics assays as well as zymogram chromogenic enzymatic staining [25]. During our research, we used ANA for zymograms on SDS–PAGE to detect esterase activity on *F. hepatica* protein extracts and linked the molecular mass detected by zymograms to its corresponding transcript as a cDNA sequence [2,3], and its molecular interaction with the catalytic serine at the serine hydrolase catalytic site is well-understood [22].

During our study, another approach was to use the ANA 3D model to locate the catalytic serine at the active site within the CestB 3D modeling. The results always indicated that serine 336 was in close proximity to the ANA ligand (Figure 2 and Figure 3); therefore, we considered it the most likely candidate to be the catalytic serine of CestB serine hydrolase. Additionally, the catalytic nature of S336 was confirmed by the consensus catalytic site pattern of serine AB hydrolase at PROSITE (prosite.expasy.org) (accessed on 10 September 2022).

The worldwide allele distribution in different hosts of the CestB amino acid substitution at position 215 suggests that the two allelic variations are equally important for the parasite, with a slight predominance of E215 over K215, with 36 and 28 percentage points of prevalence, respectively (Table 3, Figure 4). We were unable to identify an allele preference or correlation showing any linkage to a host species or country from which the parasites were obtained; however, the Sanger automatic sequence of PCR amplicons was able to detect a heterozygous condition at SNP 643, assigning an R that represents either a G or an A in equal proportions at nucleotide position 643 (Table 3, Figure 4). The fact that samples from egg-free portions from individual adult parasites presented both alleles demonstrates that heterozygous individuals exist [27]. Previous studies showed an exhaustive search of *F. hepatica* complete genomes to determine if there are multiple copies of CestB, but concluded that only one copy of CestB exists for every *F. hepatica* set of chromosomes [3]. However, the liver fluke is a diploid organism; therefore, we expect there are heterozygous individuals who transmit the different alleles present in the population according to the mechanisms of Mendelian and population genetics [28].

CestB was identified as an inducible adult *F. hepatica* carboxylesterase during TCBZ treatment of experimental parasitized sheep [2]. The gene sequence was determined from a *F. hepatica* transcriptome, and the basic protein and enzymatic properties established by GOEA and KEGG describe it as a drug-metabolizing enzyme [3]. Previous studies found that anthelmintics or their metabolites were probable targets for CestB; however, during this study, we found no connection or relationship to a particular drug in the sets of drug-like ligand affinity identified by the algorithms. There remains a possibility that there is a limited set of available ligands, and we ignored a possible interaction of CestB with TCBZ secondary metabolites. This anthelmintic compound was designed to be oxidized by monoaminoxidases to become toxic to the parasite [5], and further chemical modifications may be achieved by other enzymes before it becomes activated and acquires anthelmintic activity. Further studies and a broader set of ligands are necessary to identify other potential substrates for CestB.

## Figures and Tables

**Figure 1 genes-13-01899-f001:**
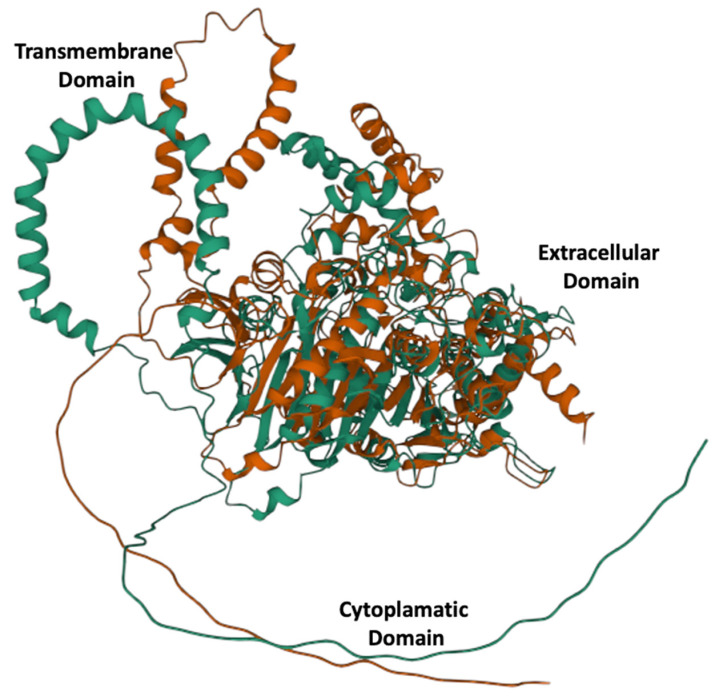
Three-dimensional alleles models for E215 green and K215 red. A significantly different structural configuration is obtained by a single radical amino acid substitution at position 215.

**Figure 2 genes-13-01899-f002:**
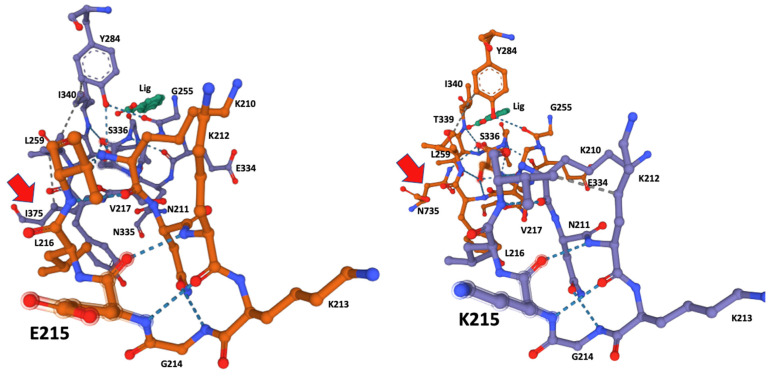
Ligand-docking modeling identifies the amino acids at the binding site in the CestB enzyme. In both alleles the amino acid at position 215 is part of the binding site and a structural reconfiguration takes place depending on the amino acid substitution, relevantly N735 is interacting via hydrogen bond with the catalytic Serine 336 in K215, (right subfigure red arrow), whereas such interaction is only partially replaced by I375 in E215 substitution (left subfigure red arrow). ANA ligand (Lig) was used to locate the position of the catalytic serine within the enzyme active site.

**Figure 3 genes-13-01899-f003:**
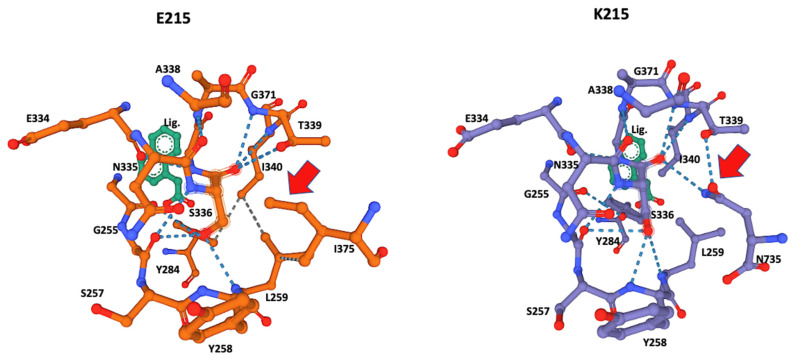
Close-up of the catalytic site 3D structure identified in CestB amino acid sequence. The S336 was identified as the catalytic serine within the active site and different configurations are detected by the two different alleles originated by substitution of amino acid 215, which according to the models, are not in close proximity to the catalytic Serine 336 and out of the structure shown in the figure; nevertheless, this substitution induces a structural reconfiguration at the core of the catalytic site, notably, the absence of the hydrogen bond interaction between N735 and the catalytic S336. The adjacent T339 in allele K215 (red arrow) is partially displaced by I375 and in the context of this allele only interacts with L259 but not with S336 (red arrow). ANA ligand (Lig) was used to locate the position of the catalytic serine within the active site.

**Figure 4 genes-13-01899-f004:**
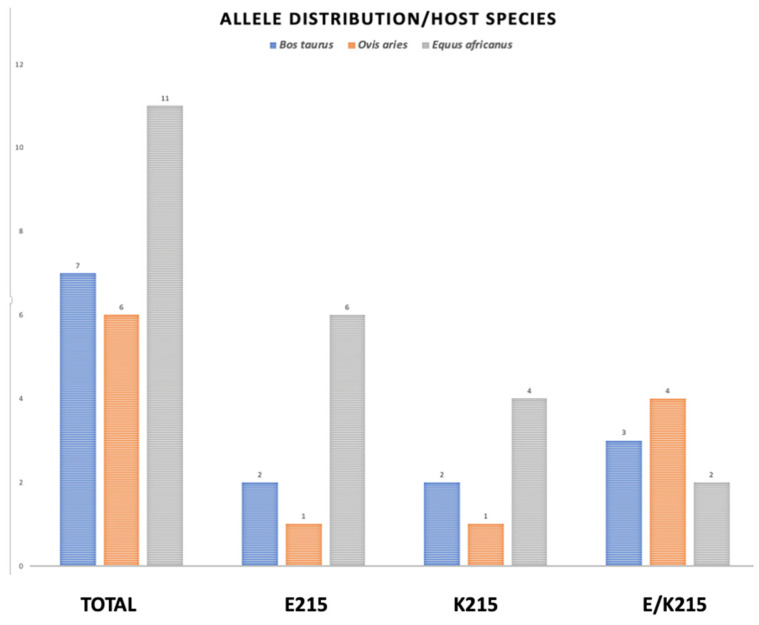
Allele distribution of amino acid substitution at position 215 in three different host species. The genomic DNA from two different developmental stages of *F. hepatica*, were submitted to PCR amplification of CestB gene and the resulting amplicons were sequenced and translated to their AA sequence and the amino acid located at position 215 determined as indicated above. The parasites obtained from *B.*
*taurus* are represented by blue bars, those obtained from *O.*
*aries* are represented by orange bars and the grey bars represent those parasites obtained from *E.*
*africanus*.

**Table 1 genes-13-01899-t001:** Protein–enzyme characteristics of CestB obtained by bioinformatics. CestB E215 allele A0A8A1L7B4 and K215 allele A0A4E0S0J7 sequences were downloaded from www.uniprot.org (accessed on 10 September 2022). Gene Ontology Enrichment analysis (GO) was obtained from geneontology.org (accessed on 10 September 2022). KEEG orthology links were obtained from www.genome.jp/kegg/ko.html (accessed on 10 September 2022). Protein domains were obtained from www.ebi.ac.uk/interpro/ (accessed on 10 September 2022).

**Cestb genbank/Uniprot Accession Numbers**	MT843326/A0A8A1L7B4
**Genbank/Uniprot Identity to other Proteins**	A0A4E0S0J7, QSS48625, THD28967, MW655750, D915_000180
**KEGG**	K03927//carboxylesterase 2 [EC:3.1.1.1 3.1.1.84 3.1.1.56]; K07378//neuroligin; K01050//cholinesterase [EC:3.1.1.8]; K07377//neurexin
**GO**	0016021//integral component of membrane
**Interpro** **Protein Domains**	AAs 141-184, 214-281 α-β hydrolase AAs 197-681 Carboxylesterase type B AAs 1-73 Cytoplasmatic Domain AAs 93-735 Extracellular Domain AAs 172-672 Carboxylesterase AAs 73-93 Transmembranal Helix AAs 74-95 Transmembranal Peptide
**Literature Description**	[2,3]

**Table 2 genes-13-01899-t002:** Ligand-binding site identification and ligand-preference prediction. CestB 3D models for both types of amino acid substitutions were analyzed using the Coach-D algorithm to find the amino acids that constitute the ligand-binding site, additionally the ligand preference prediction was obtained during the analysis and summarized.

AA Substitution	E215	K215
Binding Site AA Prediction	K210, N211, G214, E215, L216, V217, G256, L259, Y284, S336, T339, I340	K210, N211, G214, K215, L216, V217, G256, Y258, L259, S336, T339, I340, N735
Predicted S336 Hydrogen Bonds to Adjacent AAs	G256, L259, Y284, T339, I340	G256, L259, Y258, T339, I340, N735
Predicted Hydrophobic Bonds at Catalytic Site	Y284-I340 I340-L259 L259-I375	N735-Y258
Ligand Affinity Prediction/Ligand description	Ethyl hydrogen propylamidophosphate/serine hydrolase inhibitor Naphthyl acetate/synthetic esterase substrate N-acetylneuraminic acid/neurone membrane component Cyclohexyl (s)-methylphosphonofluoridate/ww2 cyclosarin Ethyl hydrogen phosphonate/serine hydrolase inhibitor 1-(2-nitrophenyl)-2,2,2-trifluoroethyl]-arsenocholine/neurotransmiter analogous Choline/neurotransmitter precursor Edrophonium\/serine hydrolase inhibitor	Ethyl hydrogen propylamidophosphate/serine hydrolase inhibitor Naphthyl acetate/synthetic esterase substrate N-ethoxyphosphonoyl-n-methyl-methanamine/serine hydrolase inhibitor Sialic acid/neurone membrane component (4r)-4-hydroxy-n,n,n-trimethylpentan-1-aminium/cholinesterase inhibitor Butyric acid/fatty acid 1-(2-nitrophenyl)-2,2,2-trifluoroethyl]-arsenocholine/cholinesterase inhibitor Ethyl hydrogen diethylamidophosphate/serine hydrolase inhibitor Tacrine/cholinesterase inhibitor Methylphosphonic acid ester/cholinesterae inhibitor Huperzine/alkaloid 3-[(1s)-1-(dimethylamino)ethyl]phenol/cholinesterase inhibitor Galanthamine/cholinesterase inhibitor

**Table 3 genes-13-01899-t003:** Summary of allelic distribution in parasites obtained from different host species and geographical origins. CestB sequences from Mexico (Mex.) and Argentina (Arg.), were obtained from parasites isolated from different host species and the SNP at position 653 as well as the amino acid substitution in position 215 were indicated. Isolates coordinates were included as location reference.

Isolate Location/Coordinates	Host Species	Parasite Stage	SNP643	AA215
MEX. 19.3026° N 98.5500 W	*Bos taurus* 1	Adult	A	K
MEX. 19.5032° N 98.5846° W	*B. taurus* 2	Eggs	R	E/K
MEX. 18.5454° N 98.2723° W	*B. taurus* 3	Adult	A	K
MEX. 20.0623° N 98.4548° W	*Ovis aries* 1	Eggs	G	E
MEX. 20.0737° N 98.4012° W	*O. aries* 2	Eggs	R	E/K
MEX. 20.1304° N 98.3459° W	*O. aries* 3	Eggs	R	E/K
MEX. 19.6861° N 98.8116° W	*B. taurus* 4	Adult	R	E/K
MEX. 19.4942° N 99.1220° W	*O. aries* 4	Eggs	R	E/K
MEX. 20.0619° N 98.2333° W	*O. aries* 5	Eggs	A	K
MEX. 18.5939° N 99.0322° W	*B. taurus* 5	Eggs	R	E/K
MEX. 18.8892° N, 99.0626° W	*B. taurus* 6	Eggs	G	E
MEX.18.8126° N, 98.9548° W	*B. taurus* 7	Eggs	G	E
MEX. 18.8995° N, 99.1733° W	*B. taurus* 8	Eggs	G	E
ARG. 33.0500° S 68.5300° W	*Equus africanus* 7	Adult	G	E
ARG. 33.0500° S 68.5300° W	*E. africanus* 8	Adult	A	K
ARG. 33.0500° S 68.5300° W	*E. africanus* 9	Adult	G	E
ARG. 33.0500° S 68.5300° W	*E. africanus* 10	Adult	G	E
ARG. 33.0500° S 68.5300° W	*E. africanus* 12	Adult	G	E
ARG. 33.0500° S 68.5300° W	*E. africanus* 13	Adult	A	K
ARG. 33.0500° S 68.5300° W	*E. africanus* 14	Adult	G	E
ARG. 33.0500° S 68.5300° W	*O. aries* 6	Adult	R	E/K
ARG. 33.0500° S 68.5300° W	*E. africanus* 15	Adult	G	E
ARG. 33.0500° S 68.5300° W	*E. africanus* 17	Adult	R	E/K
ARG. 33.0500° S 68.5300° W	*E. africanus* 19	Adult	R	E/K
ARG. 33.0500° S 68.5300° W	*E. africanus* 21	Adult	A	K

## Data Availability

Access to data of the sequences are provided as links within the manuscript in the form of raw sequencing readings, as well as GENBANK and Uniprot accession numbers for each sequence, details of raw read generated, assembly and annotation information, overall transcriptomic annotation information such as mapping rate, number of known and unknown transcript identified, splicing events and long noncoding RNA transcripts as well as the annotated gene ontology divided in number of genes found as cellular components or fulfilling a biological process or molecular function.

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
