# Peer review of "A Single Nucleotide Polymorphism Translates into a Radical Amino Acid Substitution at the Ligand-Binding Site in Fasciola hepatica Carboxylesterase B"

_genes, 2022, doi:10.3390/genes13101899_

Round 1
Reviewer 1 Report
Abstract
this section seems to short. Please add 1-2 lines to give background and aim of study followed by brief methadology, findings and conclusion.
Please add full amino acid names rather than "E" and "K".
Introduction
is appropriate. Please have a look at the reference citation in the text. line 58 shows [9][10].... it should be [9, 10] i guess.
Methods
I will suggest to first describe the study animals and their isolation and then describe the protein download and methadology.
2.6. Amplicons Visualization ,,, i think, you should remove such information from manuscript as it is too general and part of routine work.
Results
For Figure 1. rather than overlapping 3D Alleles models for E215 and K215, show them seperately so that the difference due to amino acid change may become visible.
Discussion
is appropriate. Please add more significance of this study. How it is beneficial for scientific community.
References
Please make sure that references are following the journal reference format.
Author Response
this section seems to short. Please add 1-2 lines to give background and aim of study followed by brief methodology, findings and conclusion.
Modifications made: The section was extended in details
Please add full amino acid names rather than "E" and "K".
Modifications made: Modifications were made as requested
Introduction
is appropriate. Please have a look at the reference citation in the text. line 58 shows [9][10].... it should be [9, 10] i guess.
Modifications made: Modifications were made as requested
Methods
I will suggest to first describe the study animals and their isolation and then describe the protein download and methadology.
Modifications made: Modifications were made as requested
2.6. Amplicons Visualization ,,, i think, you should remove such information from manuscript as it is too general and part of routine work.
Modifications made: The paragraph was removed as requested
Results
For Figure 1. rather than overlapping 3D Alleles models for E215 and K215, show them seperately so that the difference due to amino acid change may become visible.
Modifications made: We tried to show separate images for both alleles but it was difficult to see the differences among them, whereas when shown overlapped 3D models the dramatic changes that occur in such a big protein by a single nucleotide polymorphism are very obvious, so we decided to leave it as before
Discussion
is appropriate. Please add more significance of this study. How it is beneficial for scientific community.
Modifications made: Modifications were made as requested
References
Please make sure that references are following the journal reference format.
Modifications made: Reference format was verified
Reviewer 2 Report
The study by Miranda-Miranda and colleagues describe the analysis of a single nucleotide polymorphism in a carboxylesterase from the liver fluke parasite Fasciola hepatica. Although this study is of potential importance to the field of drug resistance in this parasite, the manuscript currently lacks sufficient background and detail regarding the results to infer the relevance and significance of these findings. The manuscript requires substantial revision before it is suitable for publication. It should also be checked by an English speaker to correct syntax and grammar.
1. Abstract – include background and relevance for this study
2. Line 22 and throughout out – the authors interchange using carboxyl esterase and carboxylesterase, be consistent with the terminology.
3. Line 40 – fasciolosis or fascioliasis, not fasciolasis
4. Line 44 – in livestock
5. Lines 46-51 – it is not currently known what the mechanism of TCBZ resistance in F. hepatica is – revise this paragraph.
6. Line 62 – in what parasite stage was this analysis?
7. Line 64-66 – what is the significance of the information in this sentence?
8. Section 2.3 -Parasite material
9. Line 91 – be more specific
10. Line 98 – how many eggs were used for the DNA as this would impact the resulting allelic variations observed?
11. Line 107
a. dNTPs
b. Also, the Promega master mix the authors used contains dNTPs, so why were more added into the reaction?
12. Line 109 – subscript 2, in H20
13. Results section need substantial editing to provide more detail of the experimental results. Further detail is required regarding the mode of activity for these types of proteins, so that the reader can make sense of the ligand modelling studies.
14. Tables 1 and 2 – need reformatting so that they are easier to interpret.
15. Line 221 and throughout – serine
16. Discussion – Although the authors have analysed the available genome data (this needs to be included in the methods and results sections), they have not analysed when the CestB is transcribed, in which life cycle stages, to infer when this carboxylesterase is important. This needs to be discussed in the context of drug resistance, as TCBZ is active against multiple life cycle stages from an early stage. Although we don’t know whether the mode of resistance is the same through the life cycle, this needs to be addressed.
Author Response
The study by Miranda-Miranda and colleagues describe the analysis of a single nucleotide polymorphism in a carboxylesterase from the liver fluke parasite Fasciola hepatica. Although this study is of potential importance to the field of drug resistance in this parasite, the manuscript currently lacks sufficient background and detail regarding the results to infer the relevance and significance of these findings. The manuscript requires substantial revision before it is suitable for publication. It should also be checked by an English speaker to correct syntax and grammar.
Modifications made: The manuscript was sent to a profesional English-editing service
1. Abstract – include background and relevance for this study
Modifications made: Additional text was added to the abstract to include context and relevance.
2. Line 22 and throughout out – the authors interchange using carboxyl esterase and carboxylesterase, be consistent with the terminology.
Modifications made: only carboxyesterase was used along the text
3. Line 40 – fasciolosis or fascioliasis, not fasciolasis
Modifications made: only fasciolosis was used along the text
4. Line 44 – in livestock
Modifications made: Modified as indicated
5. Lines 46-51 – it is not currently known what the mechanism of TCBZ resistance in F. hepatica is – revise this paragraph.
Modifications made: The paragraph was revised and modified accordingly
6. Line 62 – in what parasite stage was this analysis?
Modifications made: Adult f. hepatica was added to the text
7. Line 64-66 – what is the significance of the information in this sentence?
Modifications made:The text was modified for clarity on the potential use of ANA chromogenic ligand in zymography and 3D modeling procedures
8. Section 2.3 -Parasite material
Modifications made: The text was modified as suggested
9. Line 91 – be more specific
Modifications made: The text was modified to include further details on the parasite’s preparation and reagents used.
10. Line 98 – how many eggs were used for the DNA as this would impact the resulting allelic variations observed?
Modifications made: The text was modified to include the number of eggs used for DNA isolation as suggested
11. Line 107
Modifications made: The text was corrected
12. Line 109 – subscript 2, in H20
Modifications made: The text was corrected as suggested
13. Results section need substantial editing to provide more detail of the experimental results. Further detail is required regarding the mode of activity for these types of proteins, so that the reader can make sense of the ligand modelling studies.
Modifications made: The text at results section was modified with more details about the carboxylesterases mode of action as suggested
14. Tables 1 and 2 – need reformatting so that they are easier to interpret.
Modifications made: Tables 1 and 2 were reformatted as suggested
15. Line 221 and throughout – serine
Modifications made: The text was corrected throughout
16. Discussion – Although the authors have analysed the available genome data (this needs to be included in the methods and results sections), they have not analysed when the CestB is transcribed, in which life cycle stages, to infer when this carboxylesterase is important. This needs to be discussed in the context of drug resistance, as TCBZ is active against multiple life cycle stages from an early stage. Although we don’t know whether the mode of resistance is the same through the life cycle, this needs to be addressed.
Modifications made: The modifications made refer to a previous paper that covered this issue
CestB is included in the supplementary transcriptome data included in reference 1 contains the differential display between adults and juveniles and they exhibit similar expression levels under TCBZ treatment, however, although important, it was beyond the scope of this manuscript we are planning to use that data in a follow-up manuscript.